# Protocol for evaluating the effects of integrating music with taekwondo training in children with autism spectrum disorder: A randomized controlled trial

**Clare C. W. Yu**[1]*, **Kam M. Mok**[2], **Emma Mak**[3], **Chun T. Au**[4], **Dorothy F. Y. Chan**[5], **Stanley Wu**[1], **Raymond C. K. Chung**[1], **Matthew C. K. Ip**[1], **Simpson W. L. Wong**[6]

**1** Department of Rehabilitation Sciences, The Hong Kong Polytechnic University, Hong Kong, Hong Kong Special Administrative Region, China, **2** School of Interdisciplinary Studies, Lingnan University, Hong Kong, Hong Kong Special Administrative Region, China, **3** Private Practice, Perth, Australia, **4** Research Institute, Translational Medicine, The Hospital for Sick Children, Toronto, Ontario, Canada, **5** Faculty of Medicine, Department of Paediatrics, The Chinese University of Hong Kong, Hong Kong, Hong Kong Special Administrative Region, China, **6** Wolfson Institute of Population Health, Center for Psychiatry and Mental Health, Queen Mary University of London, London, United Kingdom

* clare-chung-wah.yu@polyu.edu.hk

## Abstract

### Background

Children with autism spectrum disorder (ASD) are known to experience difficulties in coordinating fine- and gross-motor movement. Previous interventional studies have reported significant effect of exercise-based intervention programs on improving motor skills and alleviating symptoms in ASD; however, researchers are yet to know why some participants experienced less improvement than others. One plausible explanation for suboptimal treatment outcomes is insufficient engagement in the exercise programs due to the children's difficulties in following instructions and performing the correct movements. In the proposed research, we will test the above speculation by manipulating the amount of music-therapy elements into a 10-week Taekwondo training program designed specifically for children with autism.

### Methods

This is a randomized controlled trial. Seventy-two children aged 7 to 9 years who meet the diagnostic criteria of ASD will be recruited and randomized into either the "Taekwondo training with elements of music therapy" group or the "Taekwondo training alone" group. In both groups, the participating children will attend Taekwondo training sessions twice a week over 10 consecutive weeks. In the "Taekwondo training with elements of music therapy" group, elements of music therapy will be incorporated into the Taekwondo training. Assessment will be conducted before the program commences, immediately after the completion of the 10-week program, and 2 months after the post-test. The outcome domains to be evaluated include the immediate effects during the exercise sessions. These effects include the level

relevant data from this study will be made available upon study completion.

**Funding:** CCWYu received the the General Research Fund awarded by the Research Grants Council of the University Grants Committee of Hong Kong (Reference number: GRF15612023). https://www.ugc.edu.hk/eng/rgc/funding_opport/grf/ CCWYu received the funding support from the Supporting Fund for GRF awardees and proposals rated 3.5 by RGC of the Hong Kong Polytechnic University (Reference number: P0045870). https://www.polyu.edu.hk/rio/about-rio/our-role/ Both funders will not play any role in the study design, data collection and analysis, decision to publish, or preparation of the manuscript.

**Competing interests:** The authors have declared that no competing interests exist.

of engagement in the exercise sessions (primary outcome), enjoyment, physical activity level, and rate of perceived exertion. Furthermore, the evaluation will also cover the overall effects of the training program on gross motor skills, Taekwondo skills performance, executive function, psychosocial functioning, and behavioral problems.

## Discussion/Findings

The findings of this study will inform strategies for the promotion of physical activity and engagement among children with ASD when participating in exercise training classes. The study will also suggest the importance of regular physical activity for the physical and mental well-being of children with ASD.

## Trial registration

**Clinical trial registration:** URL: http://www.clinicaltrials.gov. ClinicalTrials.gov Identifier: NCT06277778. Registered on 15 February 2024.

## Introduction

ASD is a developmental disability characterized by persistent impairments in social interaction, communication, and behavioral functioning [1]. Studies in Asia, Europe, and North America have identified an average ASD prevalence of 1.5% in their populations and have projected an increasing trend [2]. In addition to the core features of autism, motor impairments are highly prevalent in children with ASD. The prevalence of gross motor skills deficits, such as reduced balance and coordination, in children with ASD is 21% to 100% [3]. In children with ASD, impaired motor skills are significantly related to social communication skill deficits [4], emotional or behavioral disturbance [4,5], as well as decreased physical activity levels [6,7]. Deficiencies in motor control and a lack of engagement in physical activities often result in inferior physical health and higher susceptibility to chronic diseases. Notably, obesity is highly prevalent among children with ASD [8].

Studies in the last two decades have demonstrated that exercise not only improves motor skills but also attention, social cognition, and psychological well-being in children with ASD [6,9]. For example, Toscano et al. reported beneficial effects on metabolic indicators, autism traits, and parent-perceived quality of life in 6-to-12-year-olds with ASD following an exercise program [10]. However, most studies have employed exercise involving repetitive body movements (e.g., running on a treadmill), which could induce boredom. Therefore, we propose the inclusion of multi-dimensional training elements to provide all-round training on different types of motor skills, which is crucial for ASD children [11]. Taekwondo training is a good option for quantifying children's skills learning, as evidenced in our preliminary studies. Past studies have indicated that Taekwondo training improves balance control and isokinetic muscle strength in children with ASD [12] as well as developmental coordination disorder [13], but its effects on Taekwondo skills learning and mental conditions of children with ASD remain unknown. Taekwondo, a combat sport with a long history, generally involves punching, kicking, sparring, breaking, self-defense techniques, and Poomsae (a combination of offensive and defensive movement sequences). Apart from the repetitive movements of the large muscle groups, the speed and accuracy of the kicks and strikes are drilled. Moreover,

mastery of Taekwondo requires strong balance and coordination, muscle strength, and a high level of tolerance, self-control, and focus on tasks.

However, children who cannot engage actively in an exercise program may experience less improvement than anticipated. Factors contributing to the lack of engagement are multifactorial. In a survey, a group of physical educators identified inattention, social impairment, emotional regulation difficulties, and inflexible adherence to routines and structure were identified as teaching challenges [14]. We speculate that the integration of music therapy into an exercise program is a potential solution to these challenges.

Music therapy is the clinical and evidence-based use of music interventions by a credentialed music therapist to accomplish individualized goals, such as addressing the physical, emotional, cognitive, and social needs of the patient. Although children with ASD may exhibit impaired social and verbal communication, their musical skills are generally intact [15]. For example, they often demonstrate good pitch and timbre processing abilities, enhanced melodic memory, and large rhythm synchronization capacity [16], as confirmed by neuroimaging studies. For example, Sharda et al. revealed that, although speech processing is impaired in children with ASD, their song perception remains intact despite the two processes being operated by similar functional neural networks [17]. Previous studies have reported that music therapy interventions are associated with increased engagement behavior [18], emotional engagement [19], joint attention (the ability to share a focus on an object or area with another person) [20], and peer interactions [21]. However, many of these studies were clinical reports, case studies, and single-group studies with low generalizability [22,23]. In another study, Sharda et al. conducted a randomized controlled trial on music-based interventions using songs and rhythm for children with ASD. The authors demonstrated that 8–12 weeks of individual music intervention improved the children's auditory and visual attention responses, social communication, and functional brain connectivity [24]. However, studies examining the effects of music during exercise training on improving attention, active engagement, and social communication in children with ASD are scarce.

Furthermore, different musical elements may have specific influences on exercise interventions for children with ASD. For example, one study documented that musical rhythmic accompaniment enhanced gross motor performance [25]. Another study reported that slow music may motivate young children with ASD to engage in vigorous exercise [26]. We speculate that if different songs are paired with specific Taekwondo movements in the training sessions, the music could serve as useful cues, allowing children to anticipate the next movement upon hearing a particular song. Our study will apply the collaborative and consultative music therapy approach, which involves training the trainers to employ music therapy principles effectively to maximize the training's effectiveness [21]. The music therapist will provide guidance and training to the Taekwondo coaches, enabling them to standardize their instructions to the children according to the rhythm of different music used in the program.

According to Karageorghis's (2016) model, musical factors (e.g., tempo, rhythm, melody, and harmony) can be considered the antecedents that interact with the moderators (e.g., sound intensity and exercise intensity) to determine an individual's response to a piece of music. The model predicts beneficial physical and psychological outcomes associated with music used in exercise and sports [27]. In line with this model, research studies have shown that integrating music into exercise promotes pleasure and diverts attention away from the unpleasant bodily sensations generated through strenuous exercise [27,28]. Moreover, the rhythm of music can trigger the firing rate of auditory neurons. This, in turn, drives the firing patterns of motor neurons, resulting in auditory-motor synchronization—a form of rhythmic entrainment [29,30]. Based on this principle [30,31], we speculate that if the music is selected

based on the desired pace and intensity of the Taekwondo training, auditory–motor synchronization during exercise may assist in improving the motor skills of children with ASD.

The insights provided by Srinivasan et al. (2014) and their conceptual framework implicate several causal links between music experiences and the development of language, communication, and social-emotional, behavioral, and motor skills in children with ASD [32]. In addition, Karageorghis's (2016) theoretical model that addresses the antecedents, moderators, and consequences of music used in the exercise and sports domain [27,33] provided us with a framework for developing a model of combined music therapy and Taekwondo training to improve the physical and mental outcomes of children with ASD (S1 Fig). The predicted immediate effects observed during the exercise sessions include increased engagement and enjoyment, reduced perceived exertion, and increased physical activity level. The predicted overall benefits of Taekwondo training with elements of music therapy include improvements in gross motor skills, Taekwondo skills performance, attention, and social communication and behaviors.

Herein, we propose to test this hypothesis by conducting a randomized controlled trial to evaluate participating children in terms of the aforementioned study outcomes of a 10-week Taekwondo training program, with the presence or absence of music.

This trial has two objectives: (1) to examine the immediate effects of music therapy integrated into Taekwondo training sessions on the engagement and enjoyment during the training sessions in children with ASD; and (2) to examine the effects of a 10-week Taekwondo training program, with the presence or absence of music therapy, on the physical and mental outcomes in children with ASD. We hypothesize that the addition of musical elements specifically designed to match the rhythm and dynamic movement of the Taekwondo training will increase active engagement during the training sessions and provide extra benefits on physical and mental outcomes for children with ASD. The domains to be evaluated include the immediate effects during the exercise sessions of engagement (primary outcome), enjoyment, physical activity levels, and rates of perceived exertion, as well as the overall effects of the training program on gross motor skills, Taekwondo skill performance, executive function, and psychosocial functioning and behavioral problems.

## Methods

### Study design

This study will be a parallel-group randomized controlled trial, conducted and reported according to international standards (CONSORT) (Figs 1 and 2) and was registered as a clinical trial with ClinicalTrial.gov (NCT06277778).

### Participants and study setting

Flyers containing information about the study will be sent to both autism support groups for parents in the community and randomly selected primary schools in Hong Kong. The parents of potential participants who are interested in the study will be invited to attend an introductory session about the project, after which they will be given an information sheet and parental written consent form. Children meeting the following eligibility criteria will be invited to participate in the study:

Inclusion Criteria:

i.   Aged 7 to 9 years, and

ii.  Clinically diagnosed with ASD by a developmental pediatrician or clinical psychologist based on the Diagnostic and Statistical Manual of Mental Disorders (fifth edition)'s criteria for ASD [1].

| | STUDY PERIOD | | | | | |
|---|---|---|---|---|---|---|
| | Enrolment | Allocation | Post-allocation | | | Close-out |
| TIMEPOINT | -$t_1$ | $t_0$ | *Baseline* | *Post-training* | *Follow-up* | $t_x$ |
| **ENROLMENT** | | | | | | |
| **Eligibility screen** | X | | | | | |
| **Informed consent** | X | | | | | |
| **Randomisation and Allocation** | | X | | | | |
| **INTERVENTIONS** | | | | | | |
| *Taekwondo training alone* | | | ●———————● | | | |
| *Taekwondo training with music therapy* | | | ●———————● | | | |
| **ASSESSMENTS** | | | | | | |
| *Engagement in the Training Sessions* | | | ●———————● | | | |
| *Physical Activity Level* | | | ●———————● | | | |
| *Enjoyment State* | | | ●———————● | | | |
| *Rate of Perceived Exertion* | | | ●———————● | | | |
| *Gross Motor Skills* | | | X | X | X | |
| *Taekwondo skills performance* | | | X | X | X | |
| *Conners' Continuous Performance Test* | | | X | X | X | |
| *Comprehensive Trail-Making Test* | | | X | X | X | |
| *Social Responsiveness Scale* | | | X | X | X | |
| *Strength and Difficulties Questionnaires* | | | X | X | X | |
| *Clinical Global Improvement score* | | | | X | X | |
| *Statistical Analysis* | | | | | | X |

**Fig 1. SPIRIT schedule of enrolment, interventions, and assessments of the study.**

Exclusion Criteria:

i. Any experience in Taekwondo training,

ii. Received music therapy within the previous 12 months,

iii. Learning difficulties and intellectual disabilities,

iv. Sensory disorders (blindness or deafness), or

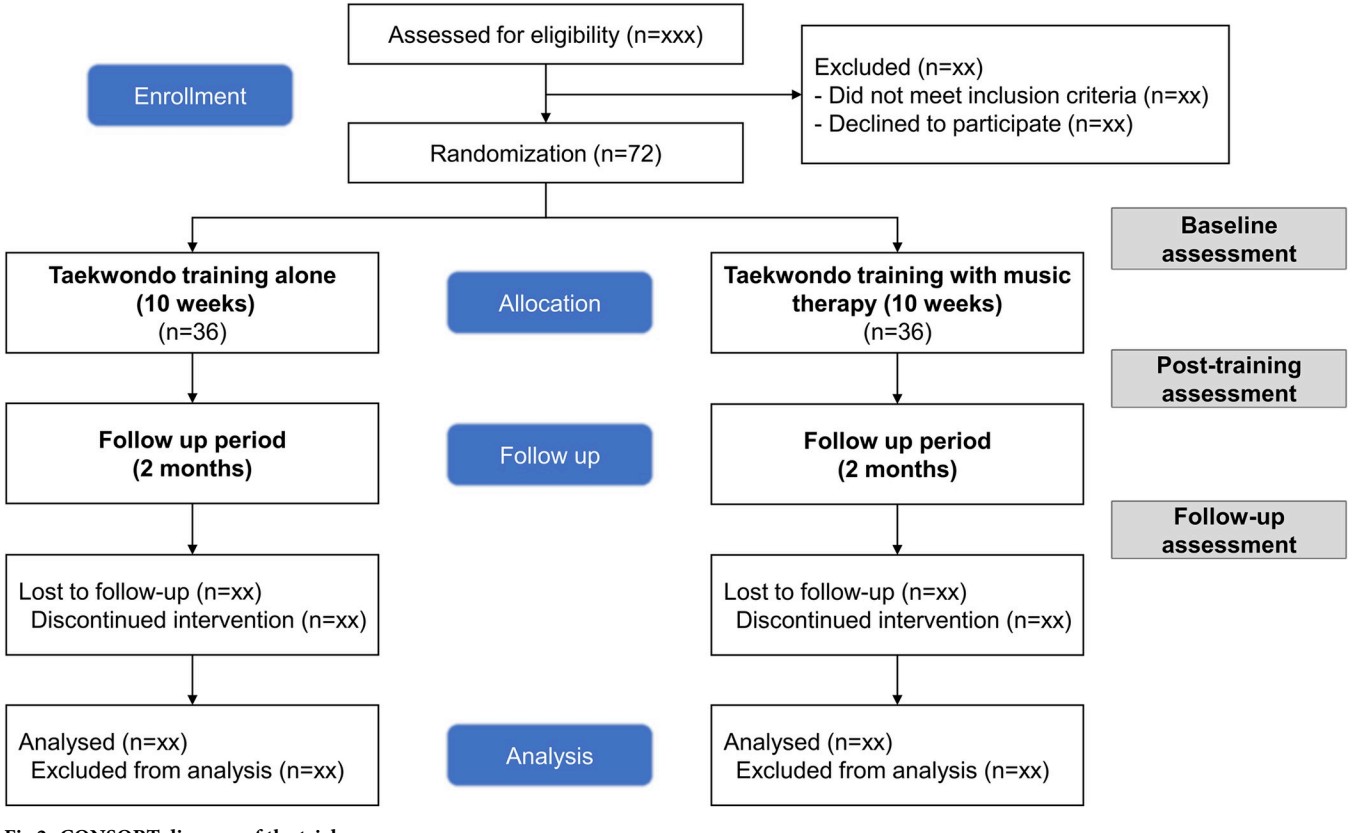

**Fig 2. CONSORT diagram of the trial.**

 v. Underlying congenital abnormalities or other diseases that limit their ability to engage in physical activities.

## Randomization and blinding

Block randomization will be implemented for this study, with varying block sizes of 2 to 4, to prevent selection bias and predictability. The randomisation list will be computer-generated and concealed by a researcher (author identifying information) who will have no direct contact with the participants. A research assistant will enroll participants and assign them to either the "Taekwondo training with elements of music therapy" (n = 36) or the "Taekwondo training alone" (n = 36) group in a 1:1 ratio. Given the behavioral nature of the intervention, blinding will not be feasible for the participants and parents. They will be unaware of their group allocation in the recruitment process until they provide written consent. The coaches, the trial coordinators, and the research personnel who review and code the video recording of the training sessions will also not be blinded to the group allocation. There will be concealment of the allocation sequence so that the trial coordinators will not know the group allocation of an individual before they consent them. To minimize measurement bias, all other research personnel involved in data collection, processing, and analysis will be blinded to group allocation.

## Intervention

In both conditions, the participating children will attend 20 sessions of Taekwondo training. Except for the experimental manipulation (the presence or absence of music), children will

receive the same level, skills, and duration of Taekwondo training across both conditions. The training will be administered in a group setting (six children per class), twice a week over 10 consecutive weeks. Each training session will last for 45 minutes and consist of warming-up exercises, hand movements, stance, foot movement techniques, a combination of stance and hand and foot techniques, and cool-down exercises. The teaching syllabus will follow the white belt (most junior) rank of the Taekwondo grading system. A qualified Taekwondo coach will conduct the training alongside a teaching assistant and two student assistants in each session.

In the Taekwondo training with elements of music therapy group, elements of music therapy will be incorporated into exercise training. Each piece of music will be tailor-made according to the rhythm, dynamic, tempo, duration, and associated feeling of a series of Taekwondo movements. For example, when forceful kicking is necessary, music with a rapid rhythm and strong, heavily accented beat will be applied. The music will serve to guide the motor patterns in a feedforward manner. The coaches will provide demonstrations that follow the tempo and rhythm of the specially composed music to guide the children in completing the Taekwondo movements. Short songs will also be composed, recorded, and played to accompany recurrent activities such as deep breathing and movement transitions. These short music pieces will offer cues to the children for recurrent activities and increase their success in effectively following the coach's lead.

Teaching assistants will document attendance rate and any significant events during the study period. Participants who have been consistently absent will be contacted to investigate the reasons for their absence and to encourage regular attendance. If a participant drops out of the training sessions, they will still be invited to complete the post-training and 2-month follow-up assessments. All kinds of ASD-related concomitant trainings are permitted during the study period and information on these trainings will be recorded.

Given the behavioral nature of the intervention, we do not anticipate that participants will have any significant adverse event during the trial. We will monitor the rate of adverse events (e.g., injury during training session) for each of the treatment arms. To maximize the safety of the training sessions, we will use a standardized Taekwondo training protocol with a trainer-to-student ratio of 1 to 2 [34].

## Study measures

Demographic data will be collected, and the children's behavioural and developmental characteristics associated with autism spectrum disorders will be evaluated at baseline using the Childhood Autism Spectrum Test (CAST) [34]. The effects of combining music with taekwondo training will be evaluated by measuring the immediate effects during the training sessions, and the overall effects of the training program on the outcome domains (Fig 3).

**1. Immediate effects in the training sessions.** *1.1 Engagement in the training sessions (primary outcome).* Training sessions will be videotaped and coded. The direct observation component of the Revised Individual Child Engagement Record (ICER-R) system will be applied to record individual children's engagement [35,36] during the training sessions. The ICER-R uses a 15-second momentary time sampling approach and involves the coding of types of engagement, the occurrence of interaction with partners (coach or peers), and whether the child was physically prompted. The types of engagement being coded at each 15-second interval are active engagement, passive engagement, active non-engagement, and passive non-engagement. Anecdotal information including emotional response was recorded during the observation for qualitative analysis. The mean inter-observer agreement is 83% for engagement, 88.7% for interaction occurrence, and 96.8% for interaction partner [35]. The engagement during the training session is expected to be the most sensitive to changes induced by

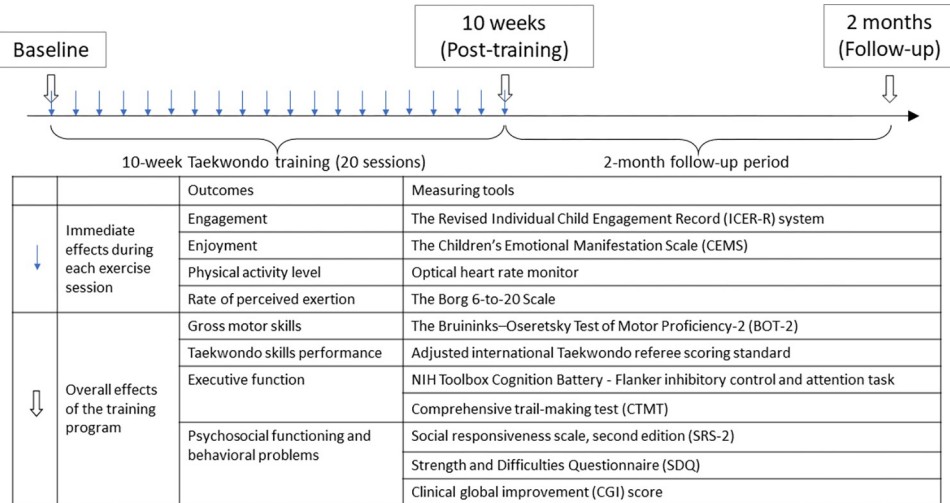

**Fig 3. Flow chart of the assessments across time.**

music therapy, as the music is specifically designed to match the rhythm and dynamics of the Taekwondo movements. This is supported by two previous studies showing that incorporating musical elements into exercise training/activities significantly increased the engagement level in children with autism [18,26].

*1.2 Enjoyment state.* The Children's Emotional Manifestation Scale (CEMS) will be used to assess the children's emotional state at the end of each training session. Children will be asked to select which cartoon facial expression most represents their emotional state. The numbers obtained for each category are added together to obtain the total score. High scores indicate the manifestation of more negative emotional behavior. The interrater reliability coefficient of the CEMS is 0.96, and its internal consistency has an alpha coefficient of 0.92 [37].

*1.3 Physical activity level.* Participants will also wear an optical heart rate monitor (Polar OH1Electro Oy, Kempele, Finland) [38] on their forearms to measure their exercise heart rate as a reflection of their physical activity level during the training. Continuous heart rate data will be recorded for every session and expressed as the mean heart rate per minute. The measured exercise heart rate greater than 55% and less than 70% of each individual's maximum heart rate (estimated as 220 minus age) will be used as individualized cutoffs for moderate- and vigorous-intensity cardiovascular workload, respectively [39].

*1.4 Rate of perceived exertion (RPE).* Children will be asked to rate their exertion level at regular time points at each training session using the Borg 6-to-20 Scale, in which "6" indicates "no exertion at all" and "20" indicates "maximal exertion" [40]. High test-retest reliability ($r = 0.89$) was found for Chinese-translated versions of the Borg 6–20 Rating of Perceived Exertion (RPE) scale, and the perceived efforts in the RPE were shown to be significantly correlated with heart rate, power output, and oxygen consumption in children ($p<0.01$) [34].

**2. Overall effects of the Taekwondo training program.** The following assessments will be conducted by an assessor experienced in the measurement tests and blinded to group allocation at baseline, after completion of the 10-week program, and at 2-month follow-up at the university laboratory and the Taekwondo training center.

*2.1 Gross motor skills.* The assessment will include gross motor composites that measure manual coordination, body coordination, and strength and agility from the Bruininks–Oseretsky Test of Motor Proficiency-2 (BOT-2) [41]. Manual coordination will be evaluated through two tasks that test upper-limb coordination. Body coordination will be assessed

through seven tasks that test bilateral coordination and nine tasks that test balance. Strength and agility will be evaluated through five tasks that assess running speed and agility and five tasks that evaluate strength. A trained assessor will demonstrate each item to the children during the assessment. Visual aids will be displayed on an easel for the children's reference. The raw scores of each task will be converted to the point scores. The point scores for each task within the same gross motor composite will be added up as a sub-score. This allows for comparisons to be made across different assessment times. The BOT-2 had good test-retest reliability and internal consistency of the total scale, with an ICC of 0.99 (95% confidence interval) and Cronbach's alpha coefficient of 0.92 [42].

*2.2 Taekwondo skills performance.* A scoring system was modified for use in this project with reference to the international Taekwondo referee scoring standard. This assessment will be carried out at the Taekwondo training center. Children's performance for a set of selected Taekwondo moves and kicks will be observed and scored by two qualified referees. Both accuracy and overall presentation of the Taekwondo movement, stance, and techniques will be scored.

*2.3 Flanker inhibitory control and attention task from the NIH Toolbox Cognition Battery app*. The NIH Toolbox Cognition Battery app is a set of clinically validated iPad-based assessments [43]. The Flanker inhibitory control and attention task is chosen and will be used to assess the inhibitory and attentional aspects of executive function. On each trial, participants are presented with five arrows (or fish for participants under eight years old) in a horizontal line. The task is to indicate the direction of the middle arrow. Twenty trials were presented, with the flanking arrows alternating randomly between congruence and incongruence with the middle arrow. A scoring algorithm integrates accuracy and reaction time, yielding scores from 0 to 10. Total administration time is about three minutes. Test-retest reliability of this task is 0.95 (95% CI 0.92, 0.97), and correlations for convergent validity is r = 0.48 [43,44].

*2.4 Comprehensive trail-making test (CTMT)*. The CTMT targets to assess visual search speed, scanning, speed of processing, mental flexibility, and executive functioning. It consists of five visual search and sequencing tasks that are heavily influenced by attention, concentration, resistance to distraction, and cognitive flexibility (or set-shifting). The child is required to connect a series of stimuli (numbers, expressed as numerals or in word form, and letters) in a specified order as fast as possible in each trial. The time scores for the five CTMT trials will be converted to standardized T-scores according to the reference tables in the testing manual, and they will be summed up to produce a global performance level score, the CTMT Composite Index (CI), which can then be converted into a percentile score according to the participant's chronological age. Higher T score and CTMT CI correspond to better performance. The CTMT has an internal consistency reliability coefficient of 0.92, and a test-retest reliability ranging from 0.70 to 0.78 [45].

*2.5 Social responsiveness scale, second edition (SRS-2)*. This scale will be completed by the parents. It is a 65-item autism rating scale widely used for identifying a spectrum of deficits in reciprocal social behavior, ranging from absent to severe, based on observations of a child's behavior in various social settings. Scoring is on a four-point Likert scale with five subscales (social awareness, social cognition, social communication, social motivation, and autistic mannerism) [46]. Higher T-scores correspond to poorer social functioning. The questionnaire has an internal consistency of 0.95 and an inter-rater reliability of 0.77 [47].

*2.6 Strength and Difficulties Questionnaire (SDQ)*. The questionnaire will be completed by parents. There are 25 items in the SDQ, which consists of five scales, each with five items, for assessing children's mental health status in terms of emotional, conduct, hyperactivity, peer problems, and prosocial behavior [48]. The questionnaire has an internal consistency coefficient of 0.81 and a test-retest reliability coefficient of 0.86 [48].

*2.7 Clinical global improvement (CGI) score.* The CGI-I will be completed by parents to evaluate children's functional changes in daily life after completion of the 10-week program, and at 2-month follow-up [49]. The CGI-I consisted of 12 questions developed by our research team, which are focused on all core symptoms of ASD and covered the following aspects: (1) motor functioning; (2) children's capability to follow instructions; (3) social, communication, and language understanding and expression; and (4) children's capabilities for attention shifting, attention to complete a task, attention to details, and imagination. The CGI-I is rated on a 7-point Likert scale ranging from 1 (very much improved) to 7 (very much worse since the initiation of intervention), with 4 representing no change. A lower score for each item of the CGI-I indicates more favorable outcomes. In addition to these 12 questions, we modified the CGI-I in this study to include a question asking parents about the overall rating of their child and an open-ended question that enables parents to report their observations of their child after participation in this program.

## Data management

All data will be encrypted, password protected, and kept in the cloud storage provided by the institution as well as principal investigator's computer. All identifiable information will be removed and replaced by study code in all the working files. A master file with study code and protected health information (name, date of birth, and contact information) will be encrypted, password protected, and kept separately. All hard-copy research documents will be locked in a cabinet. Only approved research team members will have access to these restricted data sets.

## Sample size calculation

A previous study that used jogging as an exercise format in a group of children with ASD aged 5 to 13 years demonstrated that the percentage of time the participants spent in vigorous exercise, which correlates with engagement, was significantly higher under the slow-music condition (a tempo of 60, which is similar to the songs developed in this proposal) compared with the non-music condition (mean difference = 11.8%, common SD = 21.6%, effect size d = 0.55) [26]. Another study also revealed that music increased the engagement of students with autism during group activities [18], as the percentage of time in class during which participants exhibited active engagement behaviors increased from 44.7% ± 16.9% to 68.8% ± 13% (effect size d = 1.61). Considering these studies, assuming an effect size (d) of at least 0.55 in the primary outcome proposed for this study (engagement) will be observed for the Taekwondo training with elements of music therapy condition in contrast to the Taekwondo training alone condition, a minimum of 54 subjects are required to detect such an effect with a power of 80% and type I error of 0.05. To account for a 20% attrition rate and the requirement of pairing-up children for practice per group, 72 participants will be recruited in total.

## Ethical approval

The study protocol was approved by the Human Subjects Ethics Subcommittee of the Hong Kong Polytechnic University (HSEARS20211117003-05) on 4 October 2023. All study procedures will be implemented in compliance with the Helsinki Declaration. Participants will enroll in the study on a voluntary basis. The research coordinator will explain the study and obtain written consent from parents before the study commences.

## Statistical analysis

The participants' characteristics and primary and secondary outcomes will be evaluated using summary statistics, including mean (SD), median (IQR), or proportions as appropriate, at

multiple time points. The primary analysis will consist of an intention-to-treat analysis comparing the primary outcome (engagement) between the two intervention groups; specifically, a linear mixed-effect model will be used to investigate the effect of intervention (Taekwondo training with vs without elements of music therapy) over time (interaction between intervention and time) on the engagement in the training sessions, with a random effect for each participant. The effects of music therapy on the secondary outcomes measured during the training sessions, including physical activity levels, enjoyment state, and the rate of perceived exertion, will be tested by independent samples t-tests. Further, we will use a linear regression model to investigate the effect of intervention on these outcomes, adjusting for the following prognostic variables: age, sex, and BMI. A log transformation will be applied for any outcomes resulting in a skewed distribution.

For the other secondary outcomes measured before and after the training sessions, a linear mixed-effect model will be used to investigate the effect of intervention over time (interaction between intervention and time), with a random effect for each participantand adjusting for the following prognostic variables: age, sex, and BMI. The analyses will also evaluate the sustained benefits of the training by assessing the effect of the intervention on the secondary outcomes at the 2-month follow-up using the statistical methods described above. The significance level will be set at $p < .05$, and all analyses will be performed using SPSS (v25.0 for Windows; IBM, Armonk, NY, USA).

## Discussion

In this project, we propose to evaluate the impact of incorporating music therapy into a 10-week Taekwondo training intervention designed specifically for children with ASD. Autism is a lifelong condition that begins in childhood and can affect the quality of life of the affected child and their family; therefore, the current project aims to design and test an intervention that promotes long-term participation. An intervention incorporating exercise and music is non-invasive, relatively inexpensive to other conventional interventions, and enjoyable. Furthermore, an abundance of evidence demonstrates that exercise and music can improve the symptoms of ASD, and the influence of exercise training on the affected children's physical and mental health will be long-lasting if they exercise regularly.

Compared to the 'Taekwondo-only' group, those in the 'Taekwondo with music therapy elements' group are expected to have greater physical activity levels, enhanced engagement and enjoyment, and lower rates of perceived exertion during the training sessions. These children are predicted to demonstrate greater improvement in physical function (gross motor skills and sports performance) and mental function (attention, social communication, and behaviors) after receiving the 20-session Taekwondo training with music therapy elements. Furthermore, some of the children who participate in this program may choose to take part in regular Taekwondo practice and pursue it as a lifelong passion. Parents will be able to observe the overall effects of the 20-session Taekwondo training under different conditions (i.e., the presence or absence of music) on their child, which will inform their future planning of their children's physical activities. Moreover, parents who observe their children's improvement during the program may relieve parental stress and improve parental efficacy.

The findings of this study will contribute an additional option for managing children with ASD and evidence-based strategies that can enhance children's active engagement in training sessions. The findings will benefit educational work and offer potential practical suggestions for teachers and health professionals to enhance teaching and training efficacy in children with ASD.

## Protocol amendments

Any necessary protocol changes will be effectively communicated and modified in the relevant parties (institutional review board, trial registry, and journal).

## Supporting information

**S1 Checklist. SPIRIT 2013 checklist: Recommended items to address in a clinical trial protocol and related documents\*.**
(DOCX)

**S1 Fig. The conceptual framework for this study.**
(TIF)

## Author Contributions

**Conceptualization:** Clare C. W. Yu, Emma Mak, Dorothy F. Y. Chan, Simpson W. L. Wong.

**Data curation:** Chun T. Au.

**Formal analysis:** Chun T. Au.

**Funding acquisition:** Clare C. W. Yu, Kam M. Mok, Dorothy F. Y. Chan, Simpson W. L. Wong.

**Investigation:** Clare C. W. Yu, Kam M. Mok, Emma Mak, Stanley Wu, Matthew C. K. Ip.

**Methodology:** Clare C. W. Yu, Kam M. Mok, Emma Mak.

**Project administration:** Clare C. W. Yu, Kam M. Mok, Emma Mak, Matthew C. K. Ip.

**Resources:** Clare C. W. Yu, Emma Mak, Stanley Wu.

**Supervision:** Clare C. W. Yu, Dorothy F. Y. Chan, Raymond C. K. Chung, Simpson W. L. Wong.

**Writing – original draft:** Clare C. W. Yu.

**Writing – review & editing:** Clare C. W. Yu, Kam M. Mok, Emma Mak, Chun T. Au, Dorothy F. Y. Chan, Stanley Wu, Matthew C. K. Ip, Simpson W. L. Wong.

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
