## [Decision Letter · Decision Letter 0]

2 Jul 2024

PONE-D-24-09357A study protocol for a randomized controlled trial comparing the effects of combined music with taekwondo training on the mental and physical condition of children with autism spectrum disorderPLOS ONE

Dear Dr. Yu,

Thank you for submitting your manuscript to PLOS ONE. After careful consideration, we feel that it has merit but does not fully meet PLOS ONE’s publication criteria as it currently stands. Therefore, we invite you to submit a revised version of the manuscript that addresses the points raised during the review process.

We look forward to receiving your revised manuscript.

Kind regards,

Tord Ivarsson, MD, PhD

Academic Editor

PLOS ONE

Journal Requirements:

"This study is being funded by the General Research Fund of the Research Grants Council of the University Grants Committee of Hong Kong (Reference number: GRF15612023). Additionally, the study is also receiving support from the Supporting Fund for GRF awardees and proposals rated 3.5 by RGC of the Hong Kong Polytechnic University (Reference number: P0045870). The funder had no role in trial design and will not have a role in the trial implementation, analysis, result interpretation, manuscript writing, and the decision to submit the manuscript for publication."

"CCWYu received the the General Research Fund awarded by the Research Grants Council of the University Grants Committee of Hong Kong (Reference number: GRF15612023). https://www.ugc.edu.hk/eng/rgc/funding_opport/grf/

CCWYu received the funding support from the Supporting Fund for GRF awardees and proposals rated 3.5 by RGC of the Hong Kong Polytechnic University (Reference number: P0045870). https://www.polyu.edu.hk/rio/about-rio/our-role/

Both funders will not play any role in the study design, data collection and analysis, decision to publish, or preparation of the manuscript."

**Additional Editor Comments:**

Thank you for submitting this interesting study proposal to the PLOS ONE. The research idea and design are good in general, for ex. inclusion and exclusion criteria. Procedures in general and safeguards on data safety.

However, some methodological points could be improved upon and clarified. Each of these points could be seen as a minor point to be revised, but taken together they are on the border of a major revision, which is my recommendation.

The reviewer is a highly accomplished and regarded expert in autism and neuropsychology, so I recommend you to study the comments and respond to each of them in a clear way. In general, we are in agreement.

I have as well read your manuscript and the points below are points that I think are especially pertinent to your proposal.

First, there are many assessments and data points stretching across the intervention and the 2 month follow-up. It is hard to get a sense of it all without a figure showing the assessments across time. Please make a graph or chart showing them all.

Second, I think you should include a 6- and 12 month follow-up (optional if you cannot afford it or need extra grants) as some effects may be maturing effects that will take time to be measurable. 2 month FU may be practical and good from the administrative angle, but I really think your efforts need a longer time frame.

Further, some effects may be inconstant and be waning across a prolonged period, and we need to be sure that methods included in treatment are not fleeting phenomena that will blow over within some months. In medicine in general such a time frame as 2 months FU would be regarded as inadequate except for some very special situations.

Third, in general all assessments need to show both adequate reliability and validity to be useful in treatment research. I'd recommmend you to see to it that all measures have both. If such psychometrics are not available, say so, and state why it still should be used.

Also, not all measures are sensitive to change. Possibly, you could point out the area where you expect most effect from your music tx, and verify that measures within that/those area/s are sensitive to change as well.

Your plan for statistical analysis seems on the conservative side. Using statistics based on Bayes theorem, you might decrease the risk of type II error. It might be worthwhile to consult a statistician with knowledge within this area.

So, in all, I think in a positive way of your coming work, and will welcome a revision.

Kind regards,

Tord Ivarsson

PLOS ONE academic editor

Reviewers' comments:

Reviewer's Responses to Questions

**Comments to the Author**

1. Does the manuscript provide a valid rationale for the proposed study, with clearly identified and justified research questions?

Reviewer #1: Yes

2. Is the protocol technically sound and planned in a manner that will lead to a meaningful outcome and allow testing the stated hypotheses?

Reviewer #1: Partly

3. Is the methodology feasible and described in sufficient detail to allow the work to be replicable?

Reviewer #1: Yes

4. Have the authors described where all data underlying the findings will be made available when the study is complete?

Reviewer #1: Yes

5. Is the manuscript presented in an intelligible fashion and written in standard English?

Reviewer #1: Yes

6. Review Comments to the Author

You may also provide optional suggestions and comments to authors that they might find helpful in planning their study.

Reviewer #1: This is a very interesting and well written study protocol. I have a few comments.

First, the title is very long, could it be shortened?

You write this is a single-blinded study. Clarify to whom the study is blinded and when.

I appreciate the broad intake of participants but what about IQ, could intellectual disability be an exclusion criteria?

What about inclusion/exclusion criteria for those that will perform the intervention?

The music that will be used in the training, will it be composed for this study or will music out in the public be used? What if the music is familiar to the participants, will that affect outcome. Perhaps something to take into consideration?

It is not clear what the baseline instruments are, the same as the outcome instruments? Perhaps a flow chart would be helpful to give the reader an overview of the protocol.

The neuropsychological tests Trail making test and CPT test are tests that will measure outcome. The reason for including them could described.

7. PLOS authors have the option to publish the peer review history of their article (what does this mean?). If published, this will include your full peer review and any attached files.

Reviewer #1: No

---

## [Author Response · Author response to Decision Letter 0]

15 Aug 2024

Dr. Ivarsson

Academic Editor

PLOS ONE

13 August 2024

Dear Dr. Ivarsson,

PONE-D-24-09357

We appreciate all the feedback from you and the reviewer regarding our study proposal. We have revised the proposal to address all the comments in a structured and consistent manner for the resubmission. A detailed response to the comments is included. We hope that you and the reviewer find our responses adequately address all the concerns.

Yours sincerely,

Clare YU Chung Wah

Assistant Professor

Department of Rehabilitation Sciences

The Hong Kong Polytechnic University

Journal Requirements

A: We confirm that our manuscript meets PLOS ONE's style requirements, including file naming requirements.

"This study is being funded by the General Research Fund of the Research Grants Council of the University Grants Committee of Hong Kong (Reference number: GRF15612023). Additionally, the study is also receiving support from the Supporting Fund for GRF awardees and proposals rated 3.5 by RGC of the Hong Kong Polytechnic University (Reference number: P0045870). The funder had no role in trial design and will not have a role in the trial implementation, analysis, result interpretation, manuscript writing, and the decision to submit the manuscript for publication."

We note that you have provided funding information that is not currently declared in your Funding Statement. However, funding information should not appear in the Acknowledgments section or other areas of your manuscript. We will only publish funding information present in the Funding Statement section of the online submission form. Please remove any funding-related text from the manuscript and let us know how you would like to update your Funding Statement.

Currently, your Funding Statement reads as follows: 

"CCWYu received the the General Research Fund awarded by the Research Grants Council of the University Grants Committee of Hong Kong (Reference number: GRF15612023). https://www.ugc.edu.hk/eng/rgc/funding_opport/grf/

CCWYu received the funding support from the Supporting Fund for GRF awardees and proposals rated 3.5 by RGC of the Hong Kong Polytechnic University (Reference number: P0045870). https://www.polyu.edu.hk/rio/about-rio/our-role/

Both funders will not play any role in the study design, data collection and analysis, decision to publish, or preparation of the manuscript."

A: Thank you and we have removed the funding-related text from the Acknowledgments Section of our manuscript. 

Additional Editor Comments:

1. There are many assessments and data points stretching across the intervention and the 2 month follow-up. It is hard to get a sense of it all without a figure showing the assessments across time. Please make a graph or chart showing them all.

A: We have included Fig. 4 to show the assessments across time. Please refer to the Study measures section, line 250. 

2. I think you should include a 6- and 12 month follow-up (optional if you cannot afford it or need extra grants) as some effects may be maturing effects that will take time to be measurable. 2 month FU may be practical and good from the administrative angle, but I really think your efforts need a longer time frame. 

Further, some effects may be inconstant and be waning across a prolonged period, and we need to be sure that methods included in treatment are not fleeting phenomena that will blow over within some months. In medicine in general such a time frame as 2 months FU would be regarded as inadequate except for some very special situations.

A: We acknowledge that a longer follow-up period might be preferable, as some secondary outcomes may require a longer time to show a noticeable and meaningful difference. However, based on our experience with previous studies, longer follow-up periods may increase the risk of concomitant intervention bias due to the likelihood that parents will seek out other activities or classes for their children once they have completed our training program. Additionally, additional funding will be needed to support a longer follow-up period. Therefore, we would keep the 2-month follow-up assessment only.

3. In general all assessments need to show both adequate reliability and validity to be useful in treatment research. I'd recommmend you to see to it that all measures have both. If such psychometrics are not available, say so, and state why it still should be used.

A: Additional information of reliability and validity of the assessments have been provided. Please refer to the Study measures section, lines 245 to 355. 

4. Also, not all measures are sensitive to change. Possibly, you could point out the area where you expect most effect from your music tx, and verify that measures within that/those area/s are sensitive to change as well.

A: Theoretically, the engagement measured during the training session is expected to be the most sensitive to changes induced by the music therapy, as the music is specifically designed to match the rhythm and dynamics of the Taekwondo movements. This is supported by two previous studies showing that incorporating musical elements into exercise training/activities significantly increased the engagement level in children with ASD (Carnahan C, et al 2009, Woodman AC, et al, 2018). (see lines 265 to 270) 

5. Your plan for statistical analysis seems on the conservative side. Using statistics based on Bayes theorem, you might decrease the risk of type II error. It might be worthwhile to consult a statistician with knowledge within this area.

A: We have consulted the statistician (Mr. Raymond C.K. Chung) and included him as one of the co-authors in this manuscript. Here is his response for this question: 

In this paper, we choose to use a conventional, frequentist approach to compare the efficacy of treatment on outcomes, though it may be a little bit conservative. The Bayesian approach is an inductive approach which is highly reliant on the choice of the prior distribution. Although this approach can summarize the information from external sources resulting in an increase in the precision with which the treatment effect parameter is estimated, the external evidence could perhaps be subjective opinions. Therefore, the conventional linear mixed model will be used for data analysis.

Reviewer #1: 

1. This is a very interesting and well written study protocol. I have a few comments.

First, the title is very long, could it be shortened?

You write this is a single-blinded study. Clarify to whom the study is blinded and when.

I appreciate the broad intake of participants but what about IQ, could intellectual disability be an exclusion criteria?

A: Thank you for the comments. We have made the following changes to address your comments:

a. We have revised the title to “Protocol for Evaluating the Effects of Integrating Music with Taekwondo Training in Children with Autism Spectrum Disorder: A Randomized Controlled Trial”.

b. We have revised the manuscript to clarify the blinding process as follows (see lines 202 to 210): “Given the behavioral nature of the intervention, blinding will not be feasible for the participants and parents. They will be unaware of their group allocation in the recruitment process until they provide written consent. The coaches, the trial coordinators, and the research personnel who review and code the video recording of the training sessions will also not be blinded to the group allocation. There will be concealment of the allocation sequence so that the trial coordinators will not know the group allocation of an individual before they consent them. To minimize measurement bias, all other research personnel involved in data collection, processing, and analysis will be blinded to group allocation.”

c. We have added intellectual disability as one of the exclusion criteria (see lines 189 to 194).

2. What about inclusion/exclusion criteria for those that will perform the intervention?

A: Children meeting the following eligibility criteria will be invited to participate in the study (see lines 183 to 193):

Inclusion Criteria: 

i) Aged 7 to 9 years, and

ii) Clinically diagnosed with ASD by a developmental pediatrician or clinical psychologist based on the Diagnostic and Statistical Manual of Mental Disorders (fifth edition)’s criteria for ASD [1].

Exclusion Criteria: 

iii) Any experience in Taekwondo training,

iv) Received music therapy within the previous 12 months,

v) Learning difficulties and intellectual disabilities, 

vi) Sensory disorders (blindness or deafness), or

vii) Underlying congenital abnormalities or other diseases that limit their ability to engage in physical activities.

3. The music that will be used in the training, will it be composed for this study or will music out in the public be used? What if the music is familiar to the participants, will that affect outcome. Perhaps something to take into consideration?

A: The music used in the training is specifically composed for this study. Each piece of music will be tailored to match the rhythm, dynamics, tempo, duration, and emotional tone of a specific type of Taekwondo movement. Different pieces of music will be assigned to different types of movements or activities during the training. These music pieces will serve as cues to remind children of the upcoming movements or activities, making it easier for them to follow the coach’s instructions and potentially improving their performance.

4. It is not clear what the baseline instruments are, the same as the outcome instruments? Perhaps a flow chart would be helpful to give the reader an overview of the protocol.

The neuropsychological tests Trail making test and CPT test are tests that will measure outcome. The reason for including them could described.

A: Based on the conceptual framework by Srinivasan et al. (2014) and the theoretical model by Karageorghis’s (2016), we hypothesize that incorporating musical elements specifically designed to match the rhythm and dynamics of the Taekwondo movement will enhance active engagement during the training sessions and provide additional benefits for both physical and mental outcomes in children with ASD. To evaluate the changes in these outcomes over the study period, these outcome assessments will be conducted at baseline, the end of the 10-week program, and 2-month follow-up. We have added a flow chart (Figure 4) to provide a clearer overview of the protocol. 

The Trail Making Test will be used to assess visual search speed, scanning, processing speed, mental flexibility, and executive functioning; this information is now included in the revised manuscript (see section 2.3, line 328-330). We decided to remove the CPT test from the protocol because the test takes 15 minutes to complete and requires each child to stay focused and respond promptly when a letter appears on the screen, which is extremely time-consuming and challenging for children with autism. Instead, we will use the Flanker inhibitory control and attention task from the NIH Toolbox Cognition Battery app, which serves a similar purpose by assessing the inhibitory and attentional aspects of executive function (see line 319-329).

---

## [Editor Report · Decision Letter 1]

27 Nov 2024

Protocol for Evaluating the Effects of Integrating Music with Taekwondo Training in Children with Autism Spectrum Disorder: A Randomized Controlled Trial

PONE-D-24-09357R1

Dear Dr. Yu,

We’re pleased to inform you that your manuscript has been judged scientifically suitable for publication and will be formally accepted for publication once it meets all outstanding technical requirements.

Kind regards,

Morgan E. Carlson, PhD

Academic Editor

PLOS ONE
---

## [Editor Report · Acceptance letter]

2 Jan 2025

PONE-D-24-09357R1 

PLOS ONE

Dear Dr. Yu, 

I'm pleased to inform you that your manuscript has been deemed suitable for publication in PLOS ONE. Congratulations! Your manuscript is now being handed over to our production team.

Kind regards, 

on behalf of

Dr. Morgan E. Carlson 

Academic Editor

PLOS ONE